# Inhibition of Fatty Acid-Binding Protein 4 Limits High-Fat-Diet-Associated Prostate Tumorigenesis and Progression in TRAMP Mice

**DOI:** 10.3390/ijms262110621

**Published:** 2025-10-31

**Authors:** Mingguo Huang, Shintaro Narita, Hiromi Sato, Yuya Sekine, Mizuki Kobayashi, Soki Kashima, Ryohei Yamamoto, Atsushi Koizumi, Taketoshi Nara, Kazuyuki Numakura, Mitsuru Saito, Hiroshi Nanjo, Takayuki Ikezoe, Tomonori Habuchi

**Affiliations:** 1Department of Urology, Akita University Graduate School of Medicine, Akita 010-8543, Japan; naritashintaro@gmail.com (S.N.); thabuchi@gmail.com (T.H.); 2Department of Hematology, Fukushima Medical University, Fukushima 960-1295, Japan; 3Department of Clinical Pathology, Akita University Graduate School of Medicine, Akita 010-8543, Japan; hnanjo@med.akita-u.ac.jp

**Keywords:** FABP4, high-fat diet, TRAMP mice, prostate cancer

## Abstract

Fatty acid-binding protein 4 (FABP4) is an important adipokine associated with inflammatory responses and metabolic regulation. Although a high-fat diet (HF) and/or HF-mediated obesity have been clearly linked to the progression of prostate cancer, with FABP4 potentially playing a critical role in this relationship, the mechanisms by which FABP4 facilitates this interaction remain unclear. After generating FABP4 knockout (FABP4^−/−^) transgenic adenocarcinoma of the mouse prostate (TRAMP) mice, it was found that FABP4^−/−^ TRAMP mice presented significantly ameliorated prostate tumorigenesis and tumor progression along with decreased body weight, protumorigenic cytokine secretion, and pan-amino acid synthesis when compared to TRAMP mice under the HF condition. Additionally, treatment with BMS309403—a chemical inhibitor of FABP4—was observed to abrogate the HF-mediated TRAMP tumor progression, along with reductions in body weight and cytokine production. Thus, FABP4 plays an essential role in the progression of HF-mediated prostate cancer through the modulation of metabolic and inflammatory pathways, providing a potential therapeutic target for prostate cancer.

## 1. Introduction

Dietary fat has been identified as a critical risk factor for the incidence and progression of prostate cancer (PCa), one of the most common types of cancer in men [1,2,3]. Several studies have shown that a high-fat diet (HF) and/or HF-mediated obesity may alter gene expression and cellular activity, as well as inducing other biological changes associated with the aggressiveness of PCa [4,5,6]. Fatty acid-binding protein 4 (FABP4) is an abundant protein in adipocytes and adipose tissues that regulates lipid flux and adipocyte differentiation [7,8]. FABP4 has also been associated with obesity and metabolic diseases, and is considered a promising therapeutic target for type 2 diabetes and atherosclerosis due to its roles in regulating metabolic and inflammatory pathways [9,10]. Moreover, increased expression of FABP4 has been reported in various types of cancer cells, including PCa, affecting tumor cell proliferation, angiogenesis, and metastasis [11,12,13]. Recent studies have shown that FABP4 translocates to the nucleus and influences cell growth and differentiation in both the normal prostate and PCa [14,15]. Considering this background, we can surmise that FABP4 plays an important role in the development of HF- and/or obesity-mediated PCa and, thus, may serve as a therapeutic target for PCa.

Metabolic reprograming is one of the major hallmarks of PCa, with tumor cells often modulating their metabolic pathways to biosynthesize nucleotides, amino acids, energy, and other insufficient materials, which enable uncontrolled tumor cell growth [16,17,18,19]. Furthermore, metabolic alterations often occur together with immunological dysfunctions to promote tumor progression, especially under HF or obesity [20,21,22].

In this study, we generated FABP4 knockout (FABP4^−/−^) TRAMP mice by crossing TRAMP mice with FABP4^−/−^ mice, and found that FABP4 knockout diminished HF-induced tumor development and progression in TRAMP mice while maintaining their body weight and modulating systemic cytokine secretions and intratumoral pan-amino acid synthesis.

## 2. Results

### 2.1. Generation of FABP4 Knockout (FABP4^−/−^) TRAMP Mice

FABP4^−/−^ TRAMP mice were generated by crossing male TRAMP mice with female FABP4^−/−^ mice. Both the *TRAMP*-positive PCR product (255 bp) and *FABP4* mutant PCR product (700 bp) were detected in the prostate and other organs of the newly generated FABP4^−/−^ TRAMP mice (Figure 1A and Appendix A). The FABP4 protein was highly expressed in the prostate and serum of TRAMP mice, and disappeared in FABP4^−/−^ TRAMP mice and FABP4^−/−^ mice (Figure 1B,C).

### 2.2. Loss of FABP4 Diminished HF-Mediated TRAMP Tumor Development and Progression

The progression of prostate tumors in FABP4^−/−^ TRAMP and TRAMP mice fed with different diets was compared. Accordingly, TRAMP mice exhibited prostatic epithelial neoplasia (PIN) at 12 weeks, adenocarcinoma at 24 weeks, invasion to lymph nodes and vessels, and metastasis to the lungs at 30 weeks of age [23]. Subsequently, 8-week-old male mice were randomly divided into a control diet (CD) group and a high-fat diet (HF) group and fed until certain time points (10, 12, 16, 20, 24, and 30 weeks; 6–12 mice per time subgroup) (Figure 2A,B and Appendix A). Among TRAMP mice, those consuming the HF presented significantly increased prostatic hyperplasia and PIN progression at 10–16 weeks of age and accelerated development of primary carcinoma and metastatic disease at age of 20–30 weeks, compared to those consuming the CD (Figure 2A,B). In contrast, no statistical differences in the incidence rate of prostatic burdens (e.g., hyperplasia, PIN, primary adenocarcinoma, and metastasis) were observed between FABP4^−/−^ TRAMP mice consuming HF or CD (Figure 2A,B). The incidence rates of prostatic tumor burdens were markedly decreased, as indicated by reductions in hyperplasia (by 25% at 10 weeks of age, *p* = 0.033), PIN (by 30% at 12 weeks of age, *p* = 0.026), primary carcinoma (by 22% at the age of 16 weeks, *p* = 0.037; by 32% at the age of 20 weeks, *p* = 0.021; and by 42% at the age of 24 weeks, *p* = 0.022, respectively), and metastatic disease (by 35% at the age of 24 weeks, *p* = 0.027; and 38% at the age of 30 weeks, *p* = 0.019, respectively), in the FABP4^−/−^ TRAMP-HF (TAF) group compared to the TRAMP-HF (TF) group (Figure 2A,B). The mean body weight was significantly higher in the TF group than in the other three diet groups from 16 to 30 weeks of age, whereas no significant differences were noted between the TAF and TAC groups (Figure 2C). The mean serum FABP4 level was significantly higher in the TF group than in the TC group at 12–30 weeks of age (Figure 2D). Notably, the histological analysis showed smaller and asymmetric morphological changes in the periprostatic adipocytes among FABP4^−/−^ TRAMP mice consuming the HF than among TRAMP mice (Appendix A).

### 2.3. Loss of FABP4 Inhibited Prostate Tumor Progression Through Alteration of ERK Expression in TRAMP Mice

TRAMP mice can develop differentiated primary adenocarcinoma at 24 weeks of age. Notably, the mean prostate weight (105.8 ± 16.4 and 156.1 ± 68.1 mg); incidence rates of primary prostate adenocarcinoma (33.4% and 63.7%), venule metastasis (22.2% and 45.4%), and lung metastasis (0% and 36.3%); and the level of periprostatic adipocyte infiltration were lower in HF-fed FABP4^−/−^ TRAMP mice than in TRAMP mice (Figure 3A,B). According to our previous study [13], the expression level of extracellular signal-regulated kinase (ERK) is associated with HF-induced PCa progression, and we found that the mean intensity score of p-ERK/ERK was significantly higher in the TF group at 24 weeks of age, when compared with the TAF group (Figure 3C).

### 2.4. Loss of FABP4 Decreased Protumorigenic Cytokine Secretion in TRAMP Mice

To further elucidate the role of the systemic inflammatory response in HF-induced TRAMP tumor progression mediated by FABP4, we performed a comprehensive cytokine array using sera derived from the 24-week-old FABP4^−/−^ TRAMP and TRAMP mice fed the HF (3 mice per group) (Figure 4A). Compared to TRAMP mice, FABP4^−/−^ TRAMP mice had lower serum levels of chemokine (C-X-C motif) ligand 13 (CXCL13, 6.6-fold), CXCL5 (4.0-fold), and matrix metalloproteinase-9 (MMP9, 8.0-fold) (Figure 4B). Given that decreased circulating levels of cytokines can affect the progression and metastasis of PCa in TRAMP mice, we examined the cytokine levels in the sera of all experimental age-based subgroups using mouse-specific ELISA kits for CXCL13 or CXCL5. Accordingly, our findings showed that the mean serum levels of CXCL13 (20, 24, and 30 weeks of age) and CXCL5 (16, 20, 24, and 30 weeks of age) were significantly lower in FABP4^−/−^ TRAMP-HF than in TRAMP-HF mice (Figure 4C,D). Therefore, we investigated the mRNA expression levels of CXC chemokine receptor 2 (*CXCR2*) and *CXCR5*, the functional receptors of CXCL13 and CXCL5; however, no significant differences in the prostate tumors were found between the FABP4^−/−^ TRAMP-HF and TRAMP-HF groups (Appendix A). In addition, the mRNA expression levels of *MMP9* in the mouse prostate tumor (20, 24, and 30 weeks of age) were significantly lower in FABP4^−/−^ TRAMP-HF than in TRAMP-HF mice (Figure 4E).

### 2.5. Chemical Inhibition of FABP4 Delayed TRAMP Tumorigenesis and Cytokine Production

Next, to examine the effects of BMS309403—a chemical inhibitor of FABP4—on HF-induced TRAMP tumorigenesis, 8-week-old TRAMP mice were randomly assigned to three groups: TRAMP-CD (n = 8), TRAMP-HF (n = 11), and HF with BMS309403 administration at 40 μg/mL (TRAMP-HF + BMS, n = 12) groups. The TRAMP-HF + BMS group received BMS309403 via dissolution in drinking water at 40 μg/mL throughout the dietary experiments. At 20 weeks of age, the TRAMP-HF + BMS group showed a lower incidence rate of prostate carcinoma than the TRAMP-HF group—at 8.4% and 36.3%, respectively (Figure 5A,B)—along with a decrease in body weight (Figure 5C). Additionally, the serum levels of CXCL13 and CXCL5 were significantly lower in the TRAMP-HF + BMS group than in the TRAMP-HF group (Figure 5D,E). Moreover, the mRNA expression levels of *MMP9* in the mouse prostate tumor were significantly lower in the FABP4^−/−^ TRAMP-HF group than in the TRAMP-HF group (Figure 5F).

### 2.6. Loss of FABP4 Decreased Amino Acid Synthesis in TRAMP Mice

Next, we conducted a comprehensive metabolome analysis using prostate tumor samples from 24-week-old mice in the TRAMP-HF and FABP4^−/−^ TRAMP-HF groups (three mice per group). We compared the tumor metabolites between the TRAMP-HF and FABP4^−/−^ TRAMP-HF mice, and found that nine essential and ten non-essential amino acids (excluding Cys) were 1.3–3.6-fold lower in the FABP4^−/−^ TRAMP-HF group than in the TRAMP-HF group (Figure 6A,B and Appendix A). These findings suggest that metabolomic alteration of functional amino acids may play a role in FABP4-mediated prostate tumor progression (Figure 6C).

## 3. Discussion

Despite several reports indicating that HF and/or HF-induced obesity enhances PCa progression through systematic inflammatory changes and modulation of the PCa microenvironment, the mechanisms underlying the observed increases in cytokine secretion have not been fully elucidated [20,22]. FABP4—one of the most critical adipokines—has been shown to be closely upregulated and involved in tumor cell dysfunction and progression [7,18,24]. In this study, FABP4 knockdown in HF-induced PCa attenuated serum levels of CXCL13, CXCL5, MMP9, and others, suggesting that FABP4 plays an important role in systemic cytokine secretion in TRAMP mice. In addition, FABP4 has been shown to be closely associated with prostate stromal cell activation and secretion of the cytokines IL-8 and IL-6 [13]. Furthermore, oral administration of BMS309403—a chemical inhibitor of FABP4—inhibited prostate tumor progression by modulating cytokine secretion and periprostatic adipocyte infiltration [13]. These findings suggest the potential for FABP4 as a target for preventing the development and progression of PCa by modulating the tumor microenvironment and inhibiting systemic cytokine secretion.

Investigating metabolic changes in PCa cells is an important step toward clarifying the mechanisms underlying its diet-induced progression [20,25]. This study first identified several metabolites involved in FABP4-related PCa progression, such as pan-amino acids—one of the main causes of PCa cell growth. Amino acids have numerous functions that are associated with tumor cell growth, such as energy and cytoskeleton production, maintenance of redox homeostasis, gene regulation, immunogenic modulation, and resistance to anti-cancer therapy [17,18,26,27,28,29]. Previous metabolome analyses have shown that increased tumor concentrations of amino acids were correlated with colon and lung tumor progression [30]. We explored the expression levels of FABP4 in TRAMP mice under HF conditions and found that tumor tissue concentrations of almost all amino acids—including non-essential and essential amino acids (except for Cys)—were lower among FABP4^−/−^ TRAMP than TRAMP mice, which was associated with reduced tumor progression. These findings suggest that metabolomic changes in amino acids are associated with FABP4-induced prostate tumor progression.

This study has some limitations. The FABP4 knockout TRAMP mice exhibited reduced serum levels of cytokines; however, the detailed mechanism underlying the mechanisms by which FABP4 alters prostatic epithelial inflammation could not be completely delineated. Notably, our study does suggest that FABP4 possibly induces inflammatory changes in the tumor microenvironment by affecting the crosstalk between cancer cells and surrounding stromal cells.

## 4. Conclusions

Knockdown and/or chemical inhibition of FABP4 ameliorated the development and progression of TRAMP tumors through the decreased production of protumorigenic cytokines and dynamic alteration of pan-amino acid metabolism under HF conditions. Thus, FABP4 and its associated signaling pathway could serve as promising targets for the chemoprevention and/or therapy of prostate cancer.

## 5. Materials and Methods

### 5.1. Generation of FABP4 Knockout TRAMP Mice

Transgenic adenocarcinoma of the mouse prostate (TRAMP, C57BL/6-Tg–TRAMP-8247Ng) mice were purchased from Jackson Laboratory (Bar Harbor, ME). FABP4 knockout (FABP4^−/−^) mice were kindly provided by Dr. Masato Furuhashi at the Sapporo Medical University [9]. All mice were maintained under specific pathogen-free (SPF) conditions, with our protocol having been approved by the Institutional Animal Care Committees of Akita University Graduate School of Medicine.

FABP4 knockout TRAMP mice (FABP4^−/−^ TRAMP) were generated by crossing the male TRAMP mice with the female FABP4^−/−^ mice. Genomic DNA was isolated from the tail biopsy of mice and genotyped using PCR analysis. The genotypes of *TRAMP* or non-*TRAMP* (wild-type, WT) were detected using PCR, with the following sequences of genotyping primers:

Transgene forward: 5′-TAC AAC TGC CAA CTG GGA TG-3′;

Transgene reverse: 5′-CAG GCA CTC CTT TCA AGA CC-3′;

Internal positive control forward: 5′-CTG TCC CTG TAT GCC TCT GG-3′;

Internal positive control reverse: 5′-AGA TGG AGA AAG GAC TAG GCT A-3′;

PCR conditions: 30 cycles of 60 s at 95 °C, 60 s at 55 °C, and 60 s at 72 °C, as described in the protocol of Jackson Laboratory.

*FABP4*^−/−^ or non-*FABP4*^−/−^ mice were detected using PCR with the following specific primers (common to both WT and mutant):

aP2-1: 5′-CAG CAC TCA CCC ACT TCT TTC AT-3′;

aP2 402F: 5′ ACA TAC AGG GTC TGG TCA TG-3′;

neo-2: 5′ ATA GGA GCC AGT CCC TTC CCGT 3′;

PCR conditions: 35 cycles of 60 s at 95 °C, 60 s at 55 °C, and 60 s at 72 °C, as previously described [9].

### 5.2. Animal Study

The institutional review board of the Akita University School of Medicine approved all animal experiments. Eight-week-old TRAMP or FABP4^−/−^ TRAMP male mice were randomly assigned to two different dietary groups: the CE-2 control diet (CD, Japan SLC) group and the HF (Test Diet (Purina Mills Test Diets, Richmond, IN, USA)) group. In the HF, 59.9% of the calories came from fats, 21.4% from carbohydrates, and 18.6% from proteins. Two to three mice were maintained per cage under SPF conditions, which were bred and held until they reached the desired time points (10, 12, 16, 20, 24, and 30 weeks; 6–12 mice per time subgroup) (Appendix A). The mice were provided food and water ad libitum. Body weight was measured weekly throughout the experiment.

To investigate the role of BMS309403 [2′-(5-ethyl-3,4-diphenyl-pyrazol-1-yl)-biphenyl-3-yloxy; BMS309403, Sigma, St Louis, MO, USA]—a chemical inhibitor of FABP4—eight-week-old male TRAMP mice were randomly assigned to three experimental groups (control diet group [TRAMP-CD, n = 8]: HF group [TRAMP-HF, n = 11]; and HF with BMS309403 group [TRAMP-HF + BMS, n = 12]). The FABP4 inhibitor BMS309403 was administered to mice in the TRAMP-HF + BMS group via dissolution in drinking water at 40 μg/mL, as described previously [13]. Body weight was measured weekly throughout the experiments. Thereafter, the mice were bred and held until the age of 20 weeks, considered the end of the experiments.

Mice were sacrificed at each time point. Individual prostate and other tissues were carefully dissected from the mice and subjected to metabolomic analysis, histopathology, and quantitative RT-PCR analysis. Blood was collected from the orbital sinus, and the serum was separated, filtered, and stored at −80 °C until further use, as described previously [31].

### 5.3. Histopathological Chemistry

The prostate and other tissues were carefully dissected from the mice and fixed in 10% formalin for 24 h. Thereafter, the samples were paraffin-embedded for hematoxylin and eosin (H&E) staining. Histological analysis of the prostate lesions and other tumor burdens were conducted as described previously [13]. Briefly, slides containing mouse tissue samples were classified and evaluated by M.H. and H.N. through a double-blind approach based on the following specifications: (0) normal tissue; (1) hyperplasia; (2) prostatic intraepithelial neoplasia (PIN); (3) primary adenocarcinoma; and (4) adenocarcinoma with several invasions and/or metastases, including venule invasion and lung metastasis. For each prostate sample, ten random fields were evaluated at 10× magnification. In each field, the most advanced histological feature was used for histological classification. The number of each type of lesion in the experimental groups was counted, and the percentages of different types of lesions were compared among the different experimental groups. Formalin-fixed, paraffin-embedded mouse prostate tissue sections were stained with anti-FABP4 (1:100, Abcam, Cambridge, UK), anti-ERK1/2 (1:100), and anri-pERK1/2 (Thr202/Tyr204, 1:100, Cell Signaling). Thereafter, the tissue sections were washed and incubated with HRP-labeled anti-mouse or anti-rabbit antibody (1:5000). The staining level of prostate cells was evaluated on a semi-quantitative scale.

### 5.4. Metabolome Analysis

Metabolome analysis was performed at Human Metabolome Technologies (HMT, Tsuruoka, Japan), as described previously [32]. Briefly, approximately 50 mg each of 24-week-old mouse prostate samples from the TRAMP-HF and FABP4^−/−^ TRAMP-HF groups (three mice per group) were accurately weighed, recorded, and thoroughly homogenized in extraction solvent containing internal standards. The metabolites were extracted in accordance with the protocol provided by HMT. Specifically, the samples were subjected to capillary electrophoresis–TOFMS (CE-TOFMS) analysis using the Agilent CE-TOFMS System (Agilent Technologies, CA, USA) for global analysis of the charged metabolites. The metabolites were identified by comparing their m/z values and relative migration times with those of metabolite standards. Quantification was performed by comparing the peak areas with the calibration curves generated using internal standardization techniques. Raw CE-TOFMS data were analyzed using the MasterHands ver.2.19.0.2 software.

### 5.5. Proteome Profiler Mouse Cytokine Array

The Proteome Profiler Mouse XL Cytokine Array kit was purchased from R&D Systems, Inc. (Minneapolis, MN, USA). This array consists of 111 different antibodies against each cytokine, spotted in duplicate onto membranes for semi-quantitative detection. Experiments were performed according to the manufacturer’s instructions. Briefly, array membranes were incubated for 1 h in a blocking buffer, then incubated with 1.0 mL of serum collected from the 24-week-old TRAMP-HF or FABP4^−/−^ TRAMP-HF mice for 24 h. The membranes were washed and incubated with biotinylated antibodies for 1 h at room temperature. Thereafter, the membranes were washed and the antigens were detected through incubation with a peroxidase-labeled streptavidin solution diluted to 1:1000 for 30 min. Proteins were detected via enhanced chemiluminescence, and signals were captured using the CCD camera system AE-9300 Ez-Capture MG (ATTO Instruments, Tokyo, Japan). The signal intensity in the array images was quantified and scored using the Densitograph software (ATTO). Fold changes among the samples were calculated and analyzed using the method provided by R&D Systems for Proteome Profiler Array Analysis.

### 5.6. Enzyme-Linked Immunosorbent Assay (ELISA)

Serum samples were obtained from the experimental mice. Serum levels of FABP4, CXCL13, and CXCL5 were measured using respective mouse-specific ELISA kits for FABP4, CXCL13, and CXCL5 (R&D Systems, Inc., Minneapolis, MN, USA) in duplicate, according to the manufacturer’s instructions.

### 5.7. Quantitative RT-PCR

Total RNA was extracted from the mouse tissues using TRIzol^®^ reagent (Invitrogen, Massachusetts, MA, USA). The following RT-PCR primers were used: *CXCR2*, forward 5′-ATGACTTCCAAGCTGGCCGTGG-3′, reverse 5′-CATAATTTCTGTTTGGCGCAGTGTGG-3′; *CXCR5*, forward 5′-GCTTTAAGGAGTTCCTGC-3′, reverse 5′-GGTAAGCCTACACTTTCCA-3′; matrix metalloproteinase-9 (*MMP9*), forward 5′-CGCGCAACGGGGACCACT-3′, reverse 5′-TGAGCACCATGGGATTGTAGC-3′; *β-actin*, forward 5′-ATCTGGCACCACACCTTCTA-3′, reverse 5′-CGTCATACTCCTGCTTGCTGATCC-3′. The PCR conditions were 90 °C for 30 s, 60 °C for 30 s, and 72 °C for 45 s. All experiments were performed in triplicate.

### 5.8. Statistical Analyses

All statistical analyses were performed using the SPSS version 12 software (IBM Japan, Tokyo, Japan). All values are presented as the mean ± standard deviation. Statistically significant differences were evaluated using an unpaired Student’s *t*-test or repeated measures analysis of variance (ANOVA) for comparison between two and three means in each experiment. Differences among three or more groups were determined using one-way ANOVA or the Kruskal–Wallis test for continuous variables. Differences were considered significant if *p* < 0.05.

## Figures and Tables

**Figure 1 ijms-26-10621-f001:**
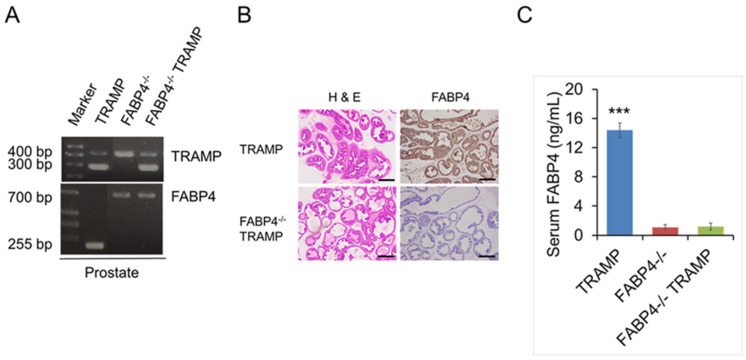
Generation of FABP4^−/−^ TRAMP mice. FABP4^−/−^ TRAMP mice were generated by crossing male TRAMP mice with female FABP4^−/−^ mice. (**A**) PCR analysis-based genotyping was performed using genomic DNA isolated from the prostates of eight-week-old TRAMP, FABP4^−/−^, and FABP4^−/−^ TRAMP mice. Both the *TRAMP*-positive PCR product (255 bp) and *FABP4* mutant PCR product (700 bp) were detected in the prostates of the newly generated FABP4^−/−^ TRAMP mice. (**B**) Prostate sections from 10-week-old TRAMP and FABP4^−/−^ TRAMP mice were stained with hematoxylin and eosin (H&E) or anti-FABP4 antibodies (1:100). bar, 100 µm. (**C**) FABP4 concentrations in sera derived from 10-week-old TRAMP, FABP4^−/−^, and FABP4^−/−^ TRAMP mice, measured using an FABP4 ELISA Kit. *** *p* < 0.001.

**Figure 2 ijms-26-10621-f002:**
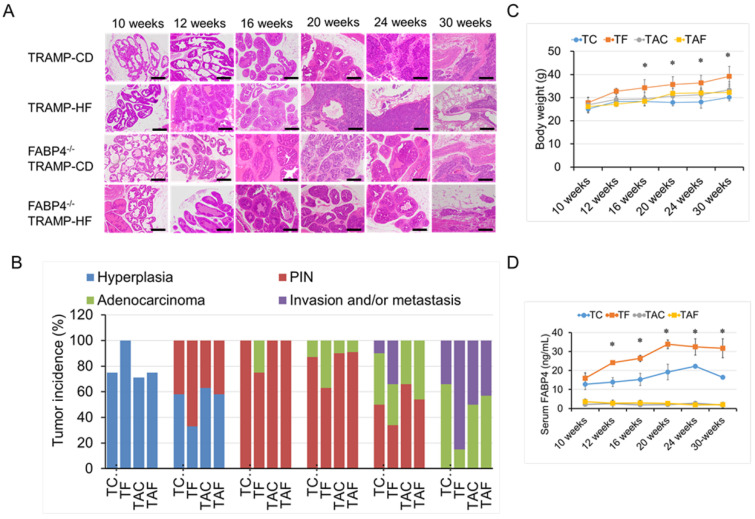
Loss of *FABP4* decreased HFD-induced PCa progression in TRAMP mice. Eight-week-old TRAMP or FABP4^−/−^ TRAMP male mice were divided into two different dietary groups—the control diet (CD) group and high-fat diet (HF) group—after which they were bred and held until they reached the desired time points (10, 12, 16, 20, 24, and 30 weeks; 6–12 mice per time subgroup). Prostate cancer progression (**A**) and incidence of prostatic burden (**B**) in the experimental mice at each time point were compared. (**A**) Unlike CD, HF induced prostatic hyperplasia and PIN progression in TRAMP mice at the age of 10–16 weeks, while also promoting primary carcinoma progression and metastatic disease at the age of 20–30 weeks. However, no differences in tumor progression were observed between the FABP4^−/−^ TRAMP-HF (TAF) and FABP4^−/−^ TRAMP-CD (TAC) groups at each time point. Bar, 100 µm. (**B**) Regarding prostatic tumor burdens, the FABP4^−/−^ TRAMP-HF (TAF) group showed decreased hyperplasia formation (25% at the age of 10 weeks, *p* = 0.033), PIN progression (30% at the age of 12 weeks, *p* = 0.026), primary carcinoma development (22% at the age of 16 weeks, *p* = 0.037; 32% at the age of 20 weeks, *p* = 0.021; and 42% at the age of 24 weeks, *p* = 0.022, respectively), and metastatic disease (35% at the age of 24 weeks, *p* = 0.027; and 38% at the age of 30 weeks, *p* = 0.019, respectively), when compared to the TRAMP-HF (TF) group. (**C**) The body weights of the experimental mice were measured weekly and compared among time point subgroups. Mean body weight was significantly higher in the TRAMP-HF (TAF) group than in the other three diet groups (ANOVA, *p* = 0.044 at 16 weeks of age; *p* = 0.029 at 20 weeks of age; *p* = 0.018 at 24 weeks of age; and *p* = 0.036 at 30 weeks of age, respectively). (**D**) Serum concentrations of FABP4 were measured using a mouse FABP4-specific ELISA kit. The serum level of FABP4 was significantly higher in the TRAMP-HF (TF) group than in the TRAMP-CD (TC) group (*p* = 0.037 at 12 weeks of age; *p*  =  0.014 at 16 weeks of age; *p*  =  0.029 at 20 weeks of age; *p*  =  0.013 at 24 weeks of age; and *p*  =  0.021 at 30 weeks of age). Baseline FABP4 levels were detected in FABP4^−/−^ TRAMP-HF (TAF) and FABP4^−/−^ TRAMP-CD (TAC) mice. * *p* < 0.05.

**Figure 3 ijms-26-10621-f003:**
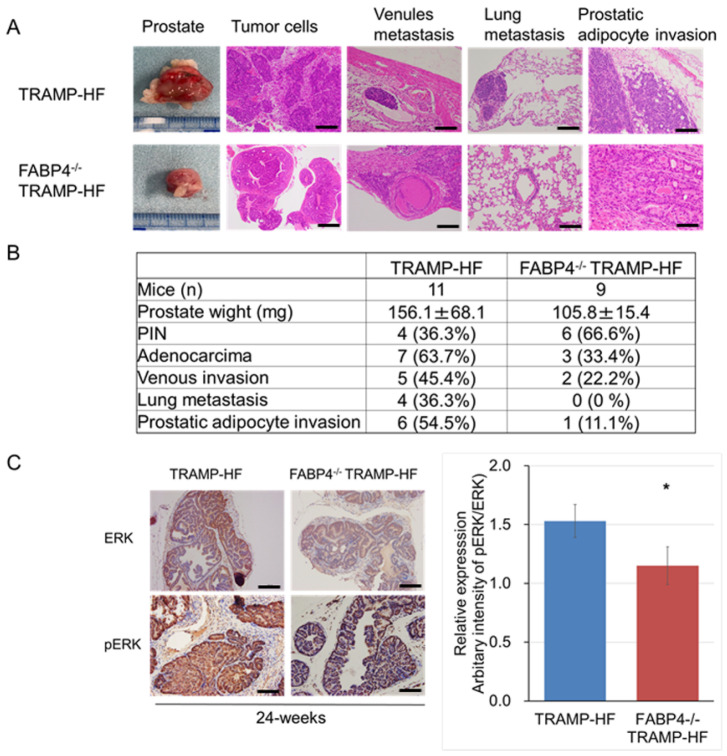
Loss of *FABP4* delayed HF-mediated prostate carcinoma development and metastasis through alteration of ERK expression in TRAMP mice. Comparison between 24-week-old TRAMP and FABP4^−/−^ TRAMP mice in terms of prostate carcinoma development and progression under HF conditions. (**A**) The mouse prostate tumor tissues were subjected to hematoxylin and eosin staining (prostate tumor cells, venule metastasis, lung metastasis, periprostatic adipocyte invasion). Bar, 100 µm. (**B**) The incidence of carcinoma development and lung and/or venule metastases was markedly lower in the FABP4^−/−^ TRAMP-HF group than in the TRAMP-HF group (*p*  =  0.025 and *p*  =  0.039, respectively). The mean weight of the prostate in the FABP4^−/−^ TRAMP-HF group was significantly lower than that in the TRAMP-HF group (*p*  =  0.034). (**C**) TRAMP tumor sections from the experimental mice were stained with anti-mouse ERK and p-ERK antibodies, with the staining intensity in the TRAMP tumor cells scored based on a semi-quantitative scale. The mean intensity score of p-ERK/ERK was significantly higher in the TRAMP-HF group than in the FABP4^−/−^ TRAMP-HF group at 24 weeks of age (1.5  ±  0.2 vs. 1.3  ±  0.2, respectively, * *p* < 0.05).

**Figure 4 ijms-26-10621-f004:**
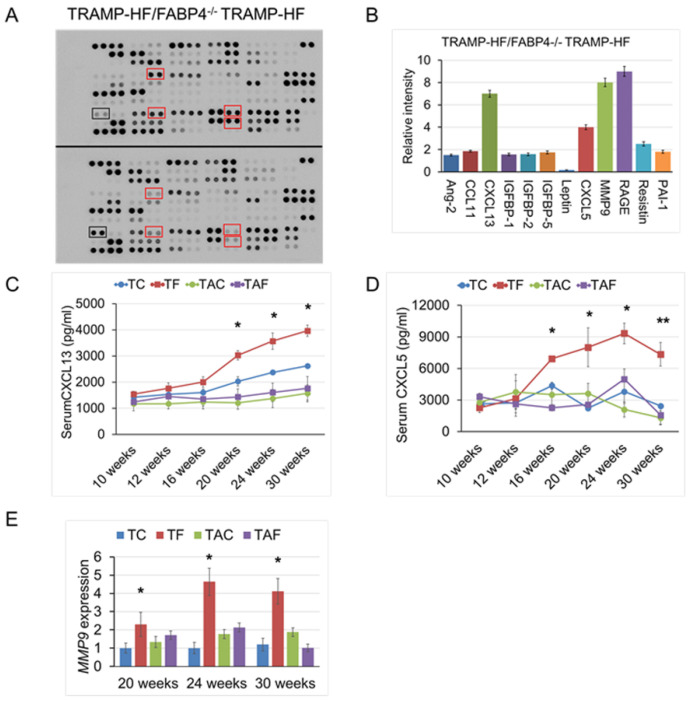
Loss of *FABP4* inhibited protumorigenic cytokine secretion in TRAMP mice. Comparison of serum cytokine levels between 24-week-old mice in the FABP4^−/−^ TRAMP-HF and TRAMP-HF groups using the comprehensive Proteome Profiler Mouse XL Cytokine Array kit (three mice per group). (**A**) The signal intensity in the array images was quantified and scored using the Densitograph software, after which fold changes were analyzed and compared between groups. (**B**) Some altered cytokine signals in the array images are indicated by rectangular boxes, such as those of CXCL13, CXCL5, MMP9, resistin, and leptin. (**C**,**D**) Serum concentrations of CXCL13 and CXCL5 in the experimental mice were measured using mouse-specific ELISA kits for CXCL13 and CXCL5. Serum levels of CXCL13 (**C**) and CXCL5 (**D**) were significantly higher in the TRAMP-HF group than in the other three groups from 16 to 30 weeks of age. ANOVA, * *p* < 0.05 and ** *p* < 0.01, respectively. (**E**) mRNA expression of *MMP9* in the prostate for each time subgroup of the experimental mice (20, 24, and 30 weeks of age) was analyzed using quantitative RT-PCR. mRNA expression levels of *MMP9* were normalized to the level of *β-actin*, and relative values were then compared among the groups. ANOVA, * *p* < 0.05.

**Figure 5 ijms-26-10621-f005:**
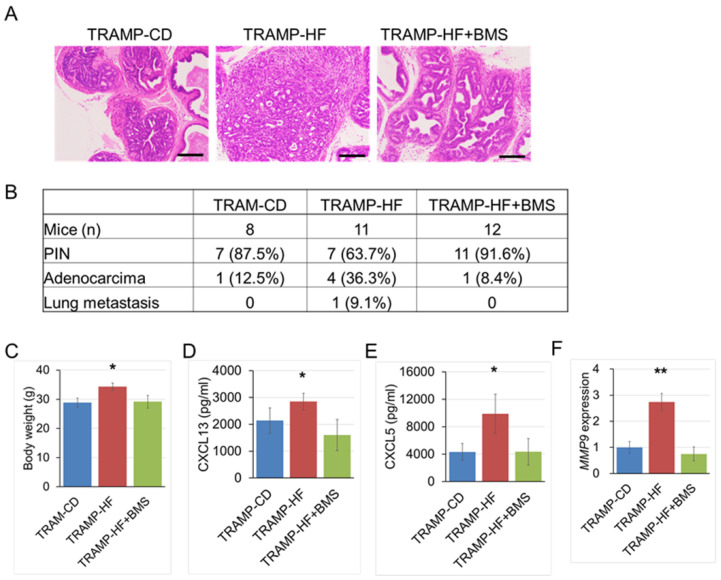
Treatment with BMS309403—a chemical inhibitor of FABP4—attenuated HF-mediated prostate cancer development in TRAMP mice. Eight-week-old male TRAMP mice were divided into three different dietary groups—a control diet (TRAMP-CD) group, a high-fat diet (TRAMP-HF) group, and an HF with BMS309403 (TRAMP-HF + BMS) group—and were then reared to the age of 20 weeks. Mice in the TRAMP-CD group were fed a CE-2 CD, whereas those in the TRAMP-HF and TRAMP-HF + BMS groups were fed an HF. Mice in the TRAMP-HF + BMS group were administered BMS309403 via dissolution in drinking water at a concentration of 40 μg/mL. Treatment with BMS309403 attenuated HF-mediated prostate cancer progression. (**A**) Slides of mouse prostate tissues subjected to hematoxylin and eosin (H&E) staining, after which the tumor burdens were evaluated. bar, 100 µm. (**B**) The incidence of primary prostate carcinoma development and metastasis were markedly lower in the TRAMP-HF + BMS group than in the TRAMP-HF group (8.4% vs. 36.3%, *p* < 0.05; and 0% vs. 9.1%, *p* < 0.05, respectively). (**C**) The mean body weight was significantly lower in the TRAMP-HF + BMS group, compared to the TRAMP-HF group. * *p* < 0.05. (**D**,**E**) Serum levels of CXCL13 (**D**) and CXCL5 (**E**) were significantly lower in the TRAMP-HF + BMS group than in the TRAMP-HF group. * *p* < 0.05. (**F**) mRNA expression levels of *MMP9* in prostate tissues were normalized to the levels of *β-actin*, after which relative values were compared among the groups. *MMP9* expression was markedly lower in the TRAMP-HF + BMS group than in the TRAMP-HF group. ** *p* < 0.01.

**Figure 6 ijms-26-10621-f006:**
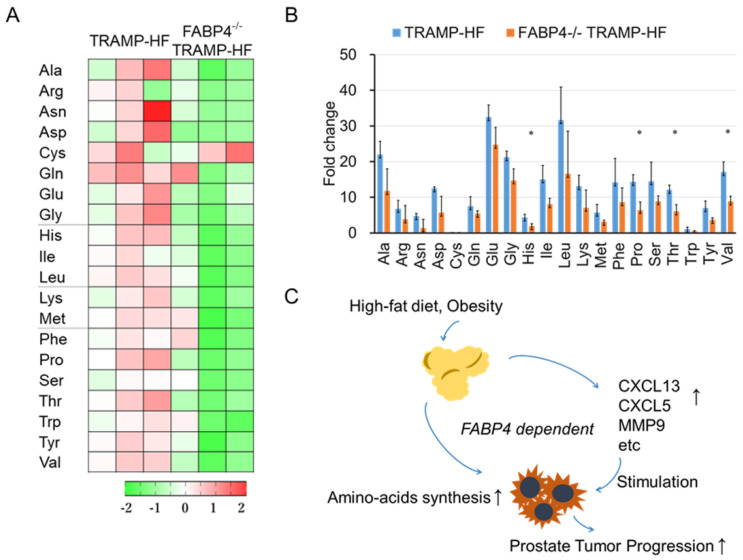
Loss of *FABP4* decreased amino acid metabolism in TRAMP mice. Prostate samples from 24-week-old mice in the TRAMP-HF and FABP4^−/−^ TRAMP-HF groups (three mice per group) were subjected to comprehensive metabolome analysis via capillary electrophoresis–TOFMS (CE-TOFMS) analysis using the Agilent CE-TOFMS System. (**A**) Cluster heat map showing relative metabolite expression of amino acids between the TRAMP-HF and FABP4^−/−^ TRAMP-HF groups. Increased levels are shown in red, while decreased levels are green. (**B**) Comparison of pan-amino acid expression between the TRAMP-HF and FABP4^−/−^ TRAMP-HF groups. The expression levels of nine essential amino acids and ten of non-essential amino acids (excluding Cys) were markedly decreased by 1.3–3.6-fold in the FABP4^−/−^ TRAMP-HF group compared with TRAMP-HF group. * *p* < 0.05. (**C**) Schematic of the putative effects of FABP4 on prostate tumor progression. HF influences PCa progression by upregulating FABP4-dependent amino acid production and protumorigenic cytokine secretion. FABP4 and its associated signaling pathway could thus serve as a promising target for the chemoprevention and/or therapy of prostate cancer.

## Data Availability

The data contributing to this research article can be accessed via the studies referenced within the article. Full details of the data are available in the article and its Appendix A. Any further information is available on request by contacting the corresponding author.

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
