# Peer review of "Inhibition of Fatty Acid-Binding Protein 4 Limits High-Fat-Diet-Associated Prostate Tumorigenesis and Progression in TRAMP Mice"

_ijms, 2025, doi:10.3390/ijms262110621_

Round 1
Reviewer 1 Report
Comments and Suggestions for Authors
Major Comments:
- The manuscript overstates the link between high-fat diet and prostate cancer, whereas some studies report mixed evidence. FABP4’s role in prostate cancer is not clearly established; citing prior cancer-related studies would improve context.
- TRAMP mice are reasonable for studying prostate cancer but do not fully recapitulate human disease. A brief justification for their use would strengthen the study.
- The manuscript lacks sample size calculations, randomization, and blinding details, limiting reproducibility and confidence in the findings.
- Some results are reported only as percentages without p-values or confidence intervals. Consistent statistical reporting is needed.
- Figures are often crowded, and legends repeat methods rather than highlighting key findings. Clearer layouts would improve readability.
- The originality is somewhat overstated because FABP4 is already linked to cancer. The novelty lies mainly in combining knockout and chemical inhibition approaches.
- Therapeutic claims are too strong given that data come from a single mouse model. The conclusions should be framed more cautiously.
Minor Comments:
- Clarify the rationale for selecting specific time points and diet groups in the methods.
- Provide more details on randomization and blinding during histological scoring.
- Simplify figure layouts and emphasize key results in the legends.
- Acknowledge limitations of the TRAMP model and discuss implications for human translation in the discussion.
- The authors should revise the text to ensure originality, properly cite all sources, and remove any unattributed content to maintain academic integrity.
References:
- Begley CG, Ellis LM. Drug development: Raise standards for preclinical cancer research.Nature. 2012;483:531–533.
- Curran-Everett D, Benos DJ. Guidelines for reporting statistics in journals published by the American Physiological Society.Am J Physiol Lung Cell Mol Physiol. 2017;313:L873–L877.
- Discacciati A, Oh JK, Wolk A. Coffee consumption and risk of non-aggressive, aggressive and fatal prostate cancer–a prospective cohort study.Am J Clin Nutr. 2012;96:878–885.
- Kilkenny C, Browne WJ, Cuthill IC, et al. Improving bioscience research reporting: The ARRIVE guidelines for reporting animal research.PLoS Biol. 2010;8:e1000412.
- Okubo H, Ando S, Ueda K, et al. Fatty acid binding protein 4 (FABP4) in the tumor microenvironment promotes prostate cancer progression.Cancer Sci. 2018;109:756–766.
- Richman EL, Kenfield SA, Chavarro JE, et al. Fat intake after diagnosis and risk of lethal prostate cancer and all-cause mortality.Am J Clin Nutr. 2012;96:855–863.
- Rougier NP, Droettboom M, Bourne PE. Ten simple rules for better figures.PLoS Comput Biol. 2014;10:e1003833.
- Shappell SB, Thomas GV, Roberts RL, et al. Prostate pathology of genetically engineered mice: definitions and classification.Toxicol Pathol. 2004;32:671–679.
- Zhang Y, Sun Y, Rao E, et al. Fatty acid-binding protein E-FABP restricts tumor growth by promoting IFN-β responses in tumor-associated macrophages.Oncotarget. 2017;8:14207–14219.
Author Response
Major Comments:
- The manuscript overstates the link between high-fat diet and prostate cancer, whereas some studies report mixed evidence. FABP4’s role in prostate cancer is not clearly established; citing prior cancer-related studies would improve context.
- TRAMP mice are reasonable for studying prostate cancer but do not fully recapitulate human disease. A brief justification for their use would strengthen the study.
- The manuscript lacks sample size calculations, randomization, and blinding details, limiting reproducibility and confidence in the findings.
- Some results are reported only as percentages without p-values or confidence intervals. Consistent statistical reporting is needed.
- Figures are often crowded, and legends repeat methods rather than highlighting key findings. Clearer layouts would improve readability.
- The originality is somewhat overstated because FABP4 is already linked to cancer. The novelty lies mainly in combining knockout and chemical inhibition approaches.
- Therapeutic claims are too strong given that data come from a single mouse model. The conclusions should be framed more cautiously.
Minor Comments:
- Clarify the rationale for selecting specific time points and diet groups in the methods.
- Provide more details on randomization and blinding during histological scoring.
- Simplify figure layouts and emphasize key results in the legends.
- Acknowledge limitations of the TRAMP model and discuss implications for human translation in the discussion.
- The authors should revise the text to ensure originality, properly cite all sources, and remove any unattributed content to maintain academic integrity.

Reviewer 2 Report
Comments and Suggestions for Authors
The manuscript by Huang et al. entitled “Inhibition of Fatty Acid Binding Protein-4 Limits the High-Fat Diet-Associated TRAMP Prostate Tumorigenesis and Progression” provides insights into the role of FABP4 in prostate cancer and the opportunity to target it pharmaceutically. The study is of interest. However, data presentation should be considerably updated.
General comments:
- English level should be significantly improved. Sometimes it is difficult to grasp the meaning of sentences.
- Gene names should be italicized
Abstract:
- Findings of the study should be summarized more specifically
- Line 24. Treatment with BMS309403
- Line 23-26. The meaning of this sentence is confusing. Please rephrase
Introduction:
- Lines 33-36. Specify metabolic or signaling pathways. In the current form, the statement is too general. In general, the links between FABP4 and obesity, as well as cardiovascular diseases and cancer are poorly described. It is suggested to further delve in specific metabolic pathways affected by altered FABP4 expression in the conditions mentioned in the Introduction. The provided evidence does not convince that “FABP4 44 plays a critical role in HF and/or obesity-mediated PCa development and may provide 45 potential therapeutic target for PCa”. Moreover, normal functions of FABP4 are not discussed.
Materials and methods:
- Protocols should be expanded to provide step-by-step instructions. Currently, this study cannot be reproduced. The strategies to quantify IHC data are not provided.
- How the total protein level was determined?
- Circulating levels of cytokines were assessed. Thus, there is no evidence that their source is the prostate gland. It is suggested to verify the findings by measuring the corresponding parameters precisely in the prostate tissue.
Results:
- Subheading 2.6 should be rephrased. It is not clear what it means “decreased amino acid metabolism”. It is suggested using “decreased amino acid synthesis”
Figures and Tables:
- Figure 1B. The quality of WB images should be improved. WB data should be quantified.
- Figures 1C, 2D, 3A, 3C and 5A. Scale bar is required.
- Figure 1D. No statistical difference is shown.
Legend should be expanded to include the key findings. In addition, legends should contain the information on the statistical data used and the way numerical values are presented (e.g., mean and SD, etc.)
Comments on the Quality of English Language- English level should be significantly improved. Sometimes it is difficult to grasp the meaning of sentences.
Author Response
Comments and Suggestions for Authors
The manuscript by Huang et al. entitled “Inhibition of Fatty Acid Binding Protein-4 Limits the High-Fat Diet-Associated TRAMP Prostate Tumorigenesis and Progression” provides insights into the role of FABP4 in prostate cancer and the opportunity to target it pharmaceutically. The study is of interest. However, data presentation should be considerably updated.
General comments:
- English level should be significantly improved. Sometimes it is difficult to grasp the meaning of sentences.
- Gene names should be italicized
Abstract:
- Findings of the study should be summarized more specifically
- Line 24. Treatment with BMS309403
- Line 23-26. The meaning of this sentence is confusing. Please rephrase
Introduction:
- Lines 33-36. Specify metabolic or signaling pathways. In the current form, the statement is too general. In general, the links between FABP4 and obesity, as well as cardiovascular diseases and cancer are poorly described. It is suggested to further delve in specific metabolic pathways affected by altered FABP4 expression in the conditions mentioned in the Introduction. The provided evidence does not convince that “FABP4 44 plays a critical role in HF and/or obesity-mediated PCa development and may provide 45 potential therapeutic target for PCa”. Moreover, normal functions of FABP4 are not discussed.
Materials and methods:
- Protocols should be expanded to provide step-by-step instructions. Currently, this study cannot be reproduced. The strategies to quantify IHC data are not provided.
- How the total protein level was determined?
- Circulating levels of cytokines were assessed. Thus, there is no evidence that their source is the prostate gland. It is suggested to verify the findings by measuring the corresponding parameters precisely in the prostate tissue.
Results:
- Subheading 2.6 should be rephrased. It is not clear what it means “decreased amino acid metabolism”. It is suggested using “decreased amino acid synthesis”
Figures and Tables:
- Figure 1B. The quality of WB images should be improved. WB data should be quantified.
- Figures 1C, 2D, 3A, 3C and 5A. Scale bar is required.
- Figure 1D. No statistical difference is shown.
Legend should be expanded to include the key findings. In addition, legends should contain the information on the statistical data used and the way numerical values are presented (e.g., mean and SD, etc.)
Comments on the Quality of English Language- English level should be significantly improved. Sometimes it is difficult to grasp the meaning of sentences.

Round 2
Reviewer 1 Report
Comments and Suggestions for Authors
None
Author Response
Reviewer2:
- We have revised the Abstract section in accordance with the reviewer’s suggestions (Page 2, line 31).
- We are thankful for the reviewer’s comment. The reviewer has mentioned that several studies (References 7–10) have previously reported the association of FABP4 with normal conditions, obesity, and metabolic diseases such as type 2 diabetes, atherosclerosis, and several types of cancers. In addition, as reported in References 11–13, FABP4 had been implicated in tumor cell proliferation, angiogenesis, and metastasis.
- The reviewer has commented about some issues regarding the details of immunohistochemistry in the mouse experiments. In the present study, the slides containing mouse tissue samples were classified and evaluated through a double-blind approach based on the following specifications: (0) normal tissue; (1) hyperplasia; (2) prostatic intraepithelial neoplasia; (3) primary adenocarcinoma; and (4) adenocarcinoma. In addition, the reviewer has pointed out the role of the circulating levels of cytokines in prostate tumorigenesis and progression. Although, the direct evidences of how the elevated serum levels of cytokines promoted prostate cancer progression in TRAMP mice could not be demonstrated in the current study, several studies have previously reported that increased serum cytokines directly and/or indirectly stimulate prostate tumor cell progression (Oncol Lett. 15(2):1607-1615, PLoS One. 19;14(12): e0226187). In addition, we have added some limitations in the Discussion section.
- The reviewer has commented regarding the phrase “decreased amino acid metabolism” in Subheading 2.6 of the Results section; however, the portions had previously been revised as per the reviewer’s suggestions in the Round1 revisions.
- We agree with the reviewer’s comment. We have incorporated scale bars in Figures 1C, 2D, 3A, 3C, and 5A, in accordance with the reviewer’s suggestion.
- We revised Figure 1C in accordance with the reviewer’s comment.
Reviewer 2 Report
Comments and Suggestions for Authors
The comments have been only partly addressed. No point-by-point responses were not provided, which made it difficult to assess how the authors addressed the comments.
The following issues are still to be addressed:
General comments:
- English level should be significantly improved. Sometimes it is difficult to grasp the meaning of sentences.
Abstract:
- Findings of the study should be summarized more specifically
Introduction:
- In general, the links between FABP4 and obesity, as well as cardiovascular diseases and cancer are poorly described. It is suggested to further delve in specific metabolic pathways affected by altered FABP4 expression in the conditions mentioned in the Introduction. The provided evidence does not convince that “FABP4 44 plays a critical role in HF and/or obesity-mediated PCa development and may provide 45 potential therapeutic target for PCa”. Moreover, normal functions of FABP4 are not discussed.
Materials and methods:
- Protocols should be expanded to provide step-by-step instructions. Currently, this study cannot be reproduced. The strategies to quantify IHC data are not provided.
- Circulating levels of cytokines were assessed. Thus, there is no evidence that their source is the prostate gland. It is suggested to verify the findings by measuring the corresponding parameters precisely in the prostate tissue.
Results:
- Subheading 2.6 should be rephrased. It is not clear what it means “decreased amino acid metabolism”. It is suggested using “decreased amino acid synthesis”
Figures and Tables:
- Scare bar is poorly visible and its size is not mentioned
- Figure 1C. No statistical difference is shown.
- English level should be significantly improved. Sometimes it is difficult to grasp the meaning of sentences.
Author Response

(The authors gave the same response as above.)

Round 3
Reviewer 2 Report
Comments and Suggestions for Authors
The comments have been addressed
Author Response
Reviewer 2:
We would like to thank the reviewer for the helpful comments and hope that the revised manuscript is now suitable for publication in International Journal of Molecular Sciences.